# Short-term Response of Greenhouse Gas Emissions from Precision Fertilization on Barley

**Carolina Fabbri \***, **Anna Dalla Marta** , **Marco Napoli** , **Simone Orlandini** **and Leonardo Verdi**

Department of Agriculture, Food, Environment and Forestry (DAGRI), University of Florence,
Piazzale delle Cascine, 18, 50144 Florence, Italy
*   Correspondence: carolina.fabbri@unifi.it

**Abstract:** Precision fertilization is a promising mitigation strategy to reduce environmental impacts of N-fertilization, but the effective benefits of variable-rate fertilization have not yet been fully demonstrated. We evaluated the short-term response (23 days) of GHGs emissions following variable-rate fertilization on barley. Yields, biomass (grains + straw) and different N-use indicators (N uptake, grain protein concentration, recovery efficiency, physiological efficiency, partial factor productivity of applied nutrient, agronomic efficiency and N surplus) were compared. Four N fertilization treatments were performed: (i) conventional– 150 kg ha$^{-1}$; (ii) variable with granular fertilizer; (iii) variable with foliar liquid supplement; (iv) no fertilization. According to proximal sensing analysis (Greenseeker Handheld) and crop needs, both variable-rate treatments accounted for 35 kg N ha$^{-1}$. Cumulative GHGs emissions were not significantly different, leading to the conclusion that the sensor-based N application might not be a GHGs mitigation strategy in current experimental conditions. Results showed that both site-specific fertilizations ensured the maintenance of high yields with a significant N rate reduction (approximately by 75%) and a N use improvement. Variable-rate N fertilization, due to similar yields (~6 tons ha$^{-1}$) than conventional fertilization and higher protein content in foliar treatment (14%), confirms its effectiveness to manage N during the later phases of growing season.

**Keywords:** carbon dioxide; methane; nitrous oxide; NDVI; proximal sensing

## 1. Introduction

The overuse of fertilizers in agriculture has been considered one of the major concerns in the public and private sectors, causing environmental pollution [1,2]. In particular, the goal of the Green Deal and European Commission is to decrease the use of nitrogen (N) by at least 20% by 2030 [3,4]. In the last few decades, the excessive use of the element led to undesirable consequences for soil (acidification process), for water (N leaching) and for atmosphere (greenhouse gasses emissions) [5]. In particular, it is reported that approximately 20–50% of the applied fertilizer is lost either as greenhouse gasses (GHGs) (e.g., methane and nitrous oxide) or other reactive N species (e.g., ammonia) [6]. N management is mainly responsible for atmospheric losses of soil organic C as carbon dioxide ($CO_2$) and methane ($CH_4$) through both increased respiration [7] and nitrous oxide ($N_2O$), with a Global Warming Potential (GWP) of 28 $CO_2$eq for $CH_4$ and 265 $CO_2$eq for $N_2O$ [8]. It is estimated that from 1% to 1.25% of the total N applied to arable soils is annually lost as $N_2O$ [9], making N fertilization of crops one of the principal sources of $N_2O$ emissions [10–12]. The analysis of different scientific reports (meta-analysis of 23 studies) showed that there is no specified dose–response effect for $N_2O$ emissions [13]. Authors reported that, regarding the formulation, nitrate ($NO_3^-$) fertilizers were predominantly responsible for $N_2O$ emissions compared to other N compounds. Generally, N fertilization also triggers $CO_2$ emissions which are, however, affected by soil characteristics, soil microbial community and fertilizer type [14,15]. Regarding $CH_4$, the emission rates appear to be influenced by N fertilization, but results are contradictory [16,17]. Understanding the dynamics of C and N pools is crucial in supporting the development of mitigation strategies in agriculture. However, in the

literature there are some inconstancies reporting that rate and time of GHGs emissions are influenced by the amount of N fertilizer applied each time and the pedoclimatic variability over the field [18,19].

In light of this, researchers and policy makers moved towards the adoption of climate-smart technologies and sustainable techniques, able to enhance crops' nitrogen use efficiency [20]. These management practices generally improve the nitrogen use efficiency (NUE) by providing better synchronization of crop N demand with N supply and have, therefore, been adopted for enhancing yield while decreasing N emissions and other losses [21]. Many studies have been recently carried out on the reliability of optical sensors used in precision farming to estimate the crop N requirements [22,23]. Due to the ability to rapidly assess the crop N content, the use of proximal sensors is a promising approach for small scale N application [24,25]. Proximal optical sensors are classified as remote sensing instruments in which the sensors are placed near the crop, indirectly assessing the crop's N status through the measurement of radiation reflectance indices [26]. The advantages of proximal sensors are that they use often their own source of energy (i.e., active sensors) minimizing the effects of ambient light conditions on reflectance readings. Moreover, they can be used any time during the growth cycle over open field or closed environments, they are not time-consuming and they can be included in fertilizer decision-making methods [27,28]. One of the most popular tools is the Greenseeker (GS; Trimble Inc., Sunnyvale, CA, USA), that has been commonly used for in-season site-specific N management [29,30]. The measurements on the crop canopy during the growth cycle provide information about the amount of N requested by the plant to achieve the same nutritional status of plants cultivated under non limiting conditions (N rich-strip), by applying a specifically developed algorithm [31]. The GS emits light in the red and near-infrared (NIR) wavelengths, used to calculate the Normalized Difference Vegetation Index (NDVI) [29]. Many studies have been carried out on the positive effect of site-specific N management on cereal yield, quality and economy for the farmer [32]. However, the GHGs emission mitigation effect of site-specific fertilization has to be investigated, due to a lack of information [33].

The aim of our study is to analyze the short-term GHGs emission dynamics by comparing different N fertilization strategies. The research is questioning if applying a suggested or conventional N rate in topdressing, there is a significant response in C and N volatilization rate in a short time after the application.

## 2. Materials and Methods

### 2.1. Experimental Design

The research was conducted at the experimental farm (WGS84, 43°47′ N; 11°13′ E, 50 m a.s.l.) of the Istituto Tecnico Agrario Statale (ITAGR), Firenze, Italy, for the growing season 2018–2019. The experiment was carried out in 12 cylindrical tanks of 1 m$^3$ (height of 90 cm) filled with soil from experimental fields of CREA ABP (Scarperia, Firenze—43°58′56″ N, 11°20′53″ E) (Table 1). The original soil profile was maintained in the tanks [34]. The tanks were of a dimension that allowed crop roots to grow unrestricted. In fact, Fan et al. [35] reported that, on average, 50% of the total root amount in barley was sited within the upper 0.12 m soil profile, 67–76% of roots can be found within the upper 0.3 m and 95% of the total root amount was accumulated within 0.99 m. Further, Poorter et al. [36] suggested that pots with a plant biomass to soil volume ratio of less than 1 g L$^{-1}$, such as in our case, had a plant biomass to soil volume ratio of the same order of magnitude as those of plants growing in the field.

Barley (*Hordeum vulgare*, L.) was sown on 25th October 2018 with a plant density of 400 seeds m$^{-2}$ and a row spacing of 0.1 m. Four N fertilization treatments were carried out in triplicate as follows: no N fertilization (Control) (0 kg N ha$^{-1}$); a locally adopted N fertilization rate (CF) (150 kg N ha$^{-1}$) with ammonium nitrate; a variable N application rate based on optical sensor measurements (GS Handheld) with ammonium nitrate (VAN); a variable N application rate based on optical sensor measurements with a foliar liquid N fertilizer supplement (VFN), from Cifo$^®$. A N rich-strip tank, with no limiting N

($200 \text{ kg ha}^{-1}$), was used as a reference to calibrate the GS. Fertilizers were distributed during the final phase of stem elongation (BBCH 39; 8 April 2019) for all treatments. Barley was manually harvested at physiological maturity on the 17 of June 2019. Daily rainfall (mm), maximum, average and minimum air temperature ($^\circ$C), and atmospheric pressure (bar), respectively, were monitored by an automatic weather station placed in the vicinity of the experimental field.

**Table 1.** Main soil physical parameters and N content at the 0–30 cm soil depth.

|  | Unit |  |
| --- | --- | --- |
| Total N | % | $0.152 \pm 0.014$ |
| Clay | % | 23.1 |
| Sand | % | 46.8 |
| Silt | % | 30.1 |
| Bulk density | g/cm$^3$ | 1.08 |

### 2.2. In-Field Estimation of Nitrogen Requirements Using Proximal Sensing

The optical sensor used in this experiment was the Greenseeker Handheld, that emits a brief burst of radiation from red (Red; $660 \pm 15$ nm) and near-infrared (NIR; $770 \pm 15$ nm) light-emitting diodes (LEDs) to accumulate reflectance data with no atmospheric disturbance. The NDVI is measured pushing a button over the device and data are registered on a liquid crystal display (LCD).

Through the GS measurements, the specific barley's N requirements at local pedoclimatic condition were assessed.

The GS measurements were performed by holding the instrument 0.6 m over the crop canopy and averaging 3 measurements per tank. The variable top-dress N rates for the VAN and VFN treatments were estimated through the N optimization algorithm proposed by [37]. The N rich-strip was used as reference for non-limiting N (NDVIref) and the Trimble Fertilization Chart was used to calculate the N rate. NDVI values were measured by means of GS on 6 April 2019. The procedure is reported in [38]. According to GS measurements and the Trimble Fertilization Chart analysis, the estimated N rate for both VAN and VFN was $35 \text{ kg N ha}^{-1}$. Matching the NDVIref with NDVI measured from the variable-rate tanks it was possible to obtain the normalized rate value. The normalized rate value was multiplied by the crop factor for barley to determine the N rate required by crops.

### 2.3. Soil GHGs Emission Measurements and Flux Estimation

Gas emissions were monitored for 23 days after fertilization, once a day for the first 5 days and once a week for the remaining period (avoiding the rainy days). An interpolation was performed to estimate the missing values from days where measurements were not performed [34]. The measurements were carried out mid-morning, when the temperatures were closer to the daily average [39]. The soil $CO_2$, $CH_4$ and $N_2O$ emissions were measured using the static chamber method and a gas analyzer XCGM 400 (Madur Sensonic) [34]. Emission fluxes were then calculated using gas concentration (ppm), chamber volume and area, molar weight of each gas and closing time. The rate of $CO_2$, $CH_4$ and $N_2O$ emissions were reported in term of carbon (C) and N per unit area (kg C ha$^{-1}$ and kg N ha$^{-1}$), respectively.

### 2.4. Crop Analysis and Nitrogen Fate within the Soil-Plant System

Crop analysis was carried out to assess the effect of different fertilization strategies in terms of yields and N uptake. Crop performances' assessment provided a cross-check on barley development and health status. At harvest, the straw and grain were separately collected from each tank and oven-dried (80 $^\circ$C; 48 h). Dried samples were weighed to determine the straw biomass (SB; kg ha$^{-1}$), grain yield (GY; kg ha$^{-1}$) and the total biomass (TB; kg ha$^{-1}$). From each tank, the N content in straw (Ns; %) and grains (Ng;

%) was determined in triplicate using a CHN analyzer (Flash EA 1112; ThermoFisher, Waltham, MA, USA). Crude protein concentration (Pc; %) was calculated by multiplying N concentration by 6.25 as reported in [40].

The main N indicators used to assess crop response to fertilization are: N uptake (Nup; kg ha$^{-1}$); Partial Factor Productivity of applied nutrient (PFP; kg grain kg$^{-1}$ N) [41]; Recovery Efficiency (RE; kg total biomass kg$^{-1}$ N), Agronomic Efficiency (AE; kg total grain kg$^{-1}$ N) and Physiological Efficiency (PE; %), (Equations (1) to (5) [42]).

$$Nup = (Ns \times SB) + (Ng \times GY) \tag{1}$$

$$PFP = GY \times Nrate^{-1} \tag{2}$$

$$RE = (Nup \text{ fertilized treatments} - Nup \text{ control}) \times Nrate^{-1} \tag{3}$$

$$AE = (GY \text{ fertilized treatments} - GY \text{ control}) \times Nrate^{-1} \tag{4}$$

$$PE = GY \times Nup^{-1} \tag{5}$$

A N surplus indicator (Nsur; kg ha$^{-1}$) was also calculated [43] (Equation (6)):

$$Nsur = Nrate - Nup \tag{6}$$

### 2.5. Statistical Analysis

Analyses of variance (ANOVAs) were performed using the R statistical program to determine treatment and N rate effects on yield, N content and N indices. All statistical comparisons were made at the $p < 0.05$ probability level unless otherwise stated. Significant differences were evaluated by a method of multiple comparisons with the Tukey honest significant difference (Tukey-HSD) test.

The Shapiro–Wilk's test was used to investigate the normality of the GHG flux data. As GHG fluxes did not show normal distribution, statistical differences between means were checked by means of the Kruskal–Wallis (K-W) test ($p < 0.05$). Then, pairwise multiple comparisons were performed by means of Dunn's post-hoc tests, with Bonferroni's $p$ value adjustment method.

## 3. Results and Discussions

### 3.1. Meteorological Conditions during the Study Period

The cumulative rainfall during the entire growth period was 460 mm. The rainfall was mainly concentrated between October and December, and between April and May, respectively, with the highest monthly cumulative rainfall in May, at the ripening stage of the barley. No rainfall occurred in January and February, while only a total of 6 mm occurred in March, corresponding to the tillering and stem elongation phases. The average temperature during the growing season was 14 °C. The coldest period occurred between December and January, followed by an increase in average daily temperatures during the spring. The highest average daily temperature was measured at the end of June (Figure 1).

### 3.2. Carbon Dioxide Fluxes

The cumulative $CO_2$ emissions for the whole period are reported in Table 2. No statistical differences were observed for the CF and VAN treatments in comparison to the Control. The VFN was significantly lower than CF and VAN on day 1 (8 April 2019), and then significantly higher than the Control on day 17 (24 April 2019) (Figure 2). The $CO_2$ emission maximum peak was observed on different days depending on the treatment. For the Control (60 kg C ha$^{-1}$), the peak was measured on the fourth day after fertilization, for the CF (91 kg C ha$^{-1}$) the peak was on the first day, while for both VFN (92 kg C ha$^{-1}$) and VAN (86 kg C ha$^{-1}$) the peak was measured on the eleventh day.

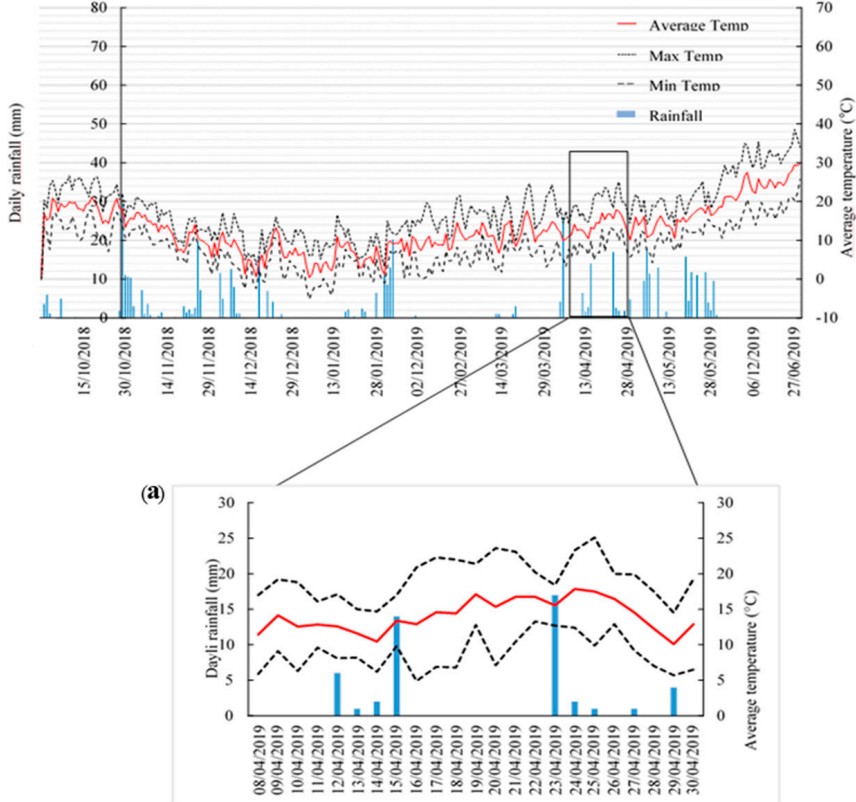

**Figure 1.** Meteorological information observed during the growing season. (**a**) indicate the emission sampling periods during the crop cycle.

**Table 2.** Cumulative emission of $CO_2$, $CH_4$ and $N_2O$ for different treatments.

| Treatment [1] | Cumulative $CO_2$ Flux (kg C ha$^{-1}$) | Cumulative $CH_4$ Flux (kg C ha$^{-1}$) | Cumulative $N_2O$ Flux (kg N ha$^{-1}$) |
|---|---|---|---|
| Control | 1066 ± 359.52 b | 5.42 ± 0.65 a | 0.08 ± 0.03 c |
| CF | 1624 ± 268.78 a | 4.22 ± 0.10 b | 0.25 ± 0.08 ab |
| VAN | 1455 ± 145.60 a | 4.52 ± 0.90 ab | 0.33 ± 0.05 a |
| VFN | 1632 ± 178.22 a | 4.22 ± 0.76 b | 0.20 ± 0.10 b |

[1] Control (no nitrogen); CF (conventional fertilization); VAN (variable-rate fertilization with foliar liquid N fertilizer); VFN (variable-rate fertilization with ammonium nitrate). Values (mean ± SE) followed by different letters within each column are significantly different at the probability level of 0.05. Different letters indicate significant difference, according to Dunn's post-hoc test ($p < 0.05$), between treatments.

Significant differences in cumulative fluxes were measured between the Control and the other treatments. Cumulative data encompassed a 23-day period as reported in Materials and Methods (9 days of actual measurements). The lowest and highest rates of $CO_2$ cumulative fluxes were registered in the Control and the VFN treatments, respectively. However, no significant difference were measured between fertilized treatments (Table 2).

The results suggested that fertilization was the driving factor in soil $CO_2$ emission dynamics regardless of the physical form of the fertilizer. Januskaitiene and Kaciene [44] showed that foliar fertilization on barley enhanced the photosynthetic rate, leading to an increase in growth and yield as a result of a positive effect on metabolic microbial activity [45]. Solid N fertilizers, from CF and VAN, produced similar cumulative $CO_2$ emissions flux to VFN still encouraging crop development and growth that were higher than the Control. Despite the higher N rate of CF, no differences were observed on $CO_2$ emissions compared to variable-rate treatments. This is probably due to an overabundance of N that inhibited soil microbial respiration limiting $CO_2$ emissions fluxes only from root respiration process [46,47].

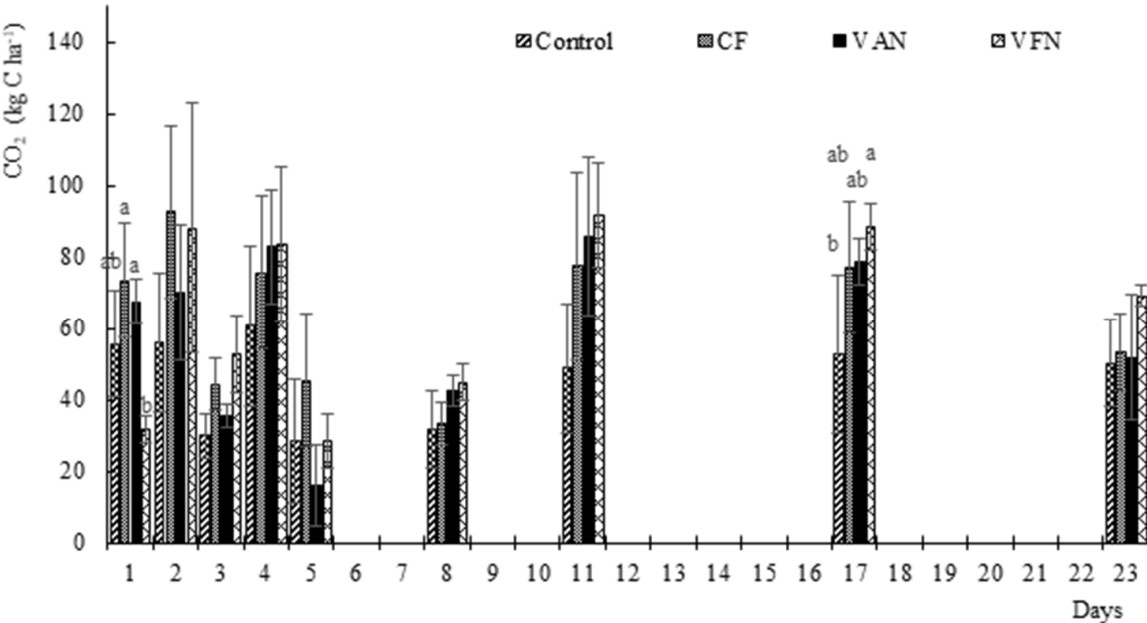

**Figure 2.** $CO_2$ emission dynamics (kg $CO_2$-C ha$^{-1}$) for barley in season 2018/2019. Different letters indicate significant difference, according to Dunn's post-hoc test ($p < 0.05$), between treatments.

In our experiment, emissions appeared to be more related to other factors than to the N rate. In general, $CO_2$ production is suggested to be mostly regulated by the interactions among vegetation type, soil temperature, soil moisture, root activity and other factors [48]. Soil temperature and moisture are assumed to be the most important drivers [49]. In line with this, $CO_2$ emission dynamics in this experiment were positively and significantly ($p < 0.05$) correlated with temperature, probably due to an increase in the decomposition of the fertilizers caused by the mineralization process [50]. Therefore, in the fertilized plots, we observed a higher emission variability compared to the Control where, from day 10 to 23 (from 17 April 2019 to 30 April 2019), the daily emissions had an almost constant rate. In the present study, the highest emission peaks for the N-fertilized plots were evident after rainfall events on day 11 (18 April 2019) and day 17 (24 April 2019) (Figure 1). These results corroborated previous studies, where the largest emission peaks were suggested to be induced by rainfall [51]. This was probably due to the respiratory activity in the soil occurring in all treatments following the degradation of organic matter and nutrients contained in the soil from residues of the previous crop [52].

### 3.3. Methane Fluxes

$CH_4$ fluxes were generally lower than $CO_2$ throughout the monitoring period (Figure 3). No statistical differences were evident in the daily emissions, showing only an independent significant ($p < 0.05$) trend for the Control compared to the VFN treatment on Day 1 (8 April 2019). For the VAN, VFN and Control treatments, the emission peak occurred on the fourth day (0.50, 0.47 and 0.56 kg C ha$^{-1}$, respectively), while for CF it was on the fifth day (0.45 kg C ha$^{-1}$).

The cumulative $CH_4$ emission fluxes for the entire period are reported in Table 2. The analysis of cumulative fluxes showed the highest $CH_4$ emissions for the Control treatment and the lowest for VFN. However, differences were not significant. In this case, temperature and rainfall during the monitoring period (Figure 1) might have been played a greater role, especially the wet soil conditions.

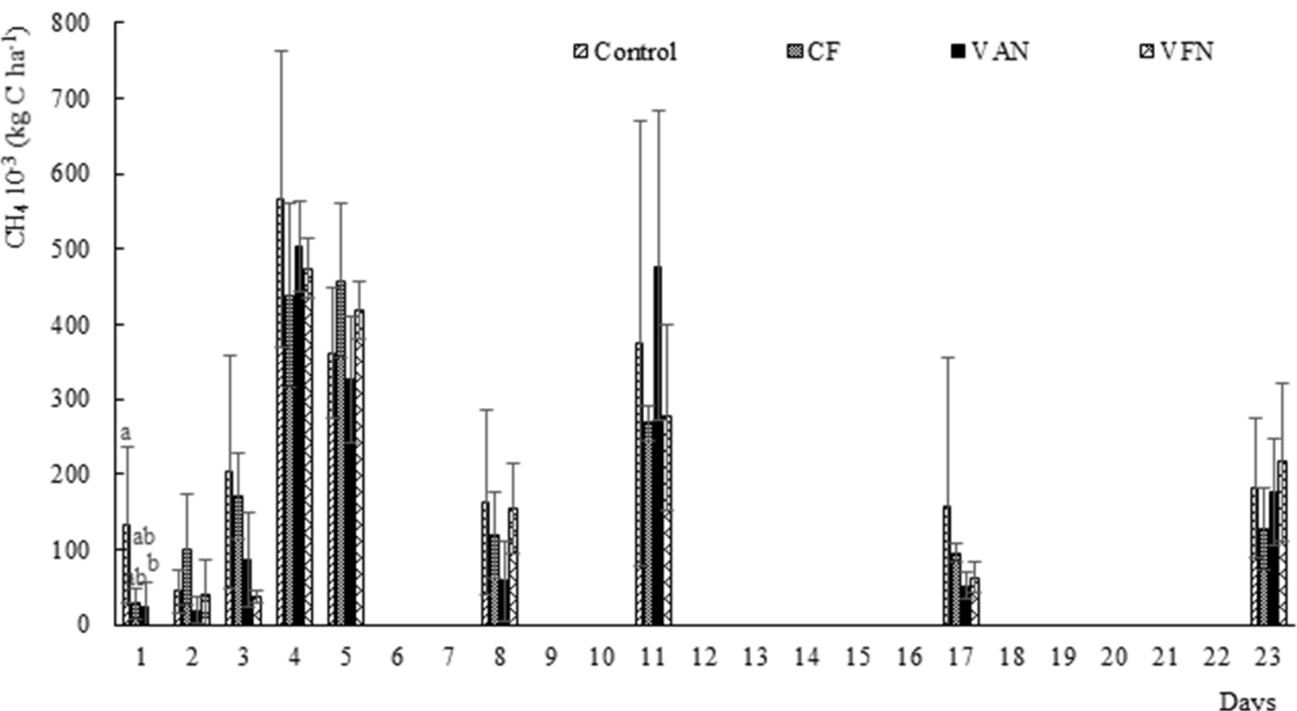

**Figure 3.** $CH_4$ emission dynamics ($CH_4$ $10^{-3}$ kg C ha$^{-1}$) for barley in season 2018/2019. Different letters indicate significant difference, according to Dunn's post-hoc test ($p < 0.05$), between treatments.

A reason for the lowest $CH_4$ flux in the N foliar fertilization might be the modification in root exudates, commonly induced by its effect on the carbohydrate spectrum [53]. These observations validate the hypothesis that $CH_4$ fluxes are not influenced by N fertilization, as was reported by [54]. Similarly to our observations, the authors of [55] reported no significant effects of N fertilization on $CH_4$ emission dynamics in a long-term experiment on barley. The higher $CH_4$ emissions for the Control compared to the CF treatment may be due to a response in $CH_4$ uptake by the fertilized plots. In fact, a positive correlation exists between $CH_4$ uptake and N fertilization [56].

Although N fertilizers may regulate $CH_4$ emissions from soils [57], our results suggested that other factors, such as plant community structure, plant litter and roots have an important influence [58]. In addition, soil $CH_4$ production requires strictly anaerobic conditions and correlates positively with soil humidity [59].

### 3.4. Nitrous Oxide Fluxes

On Day 1 (8 April 2019) and 3 (10 April 2019) a significant difference in $N_2O$ flux was observed among treatments, having the highest emissions for VAN (0.079 $10^{-3}$ kg N ha$^{-1}$) on the first day and the highest emissions for VFN (0.062 $\times$ $10^{-3}$ kg N ha$^{-1}$) on the third day (Figure 4). From day 11 (18 April 2019), the VAN showed the highest emission rate that was significantly higher than the Control and CF, respectively. The VAN was also significantly different on Day 17 from the VFN (0.004 $\times$ $10^{-3}$ kg N ha$^{-1}$). At the end of the sampling period, a general reduction in emissions was observed. $N_2O$ emissions from the Control were produced only at the beginning of the sampling period, for 4 days, and this is probably due to the intrinsic soil N content (Table 1). The cumulative $N_2O$ emission fluxes for the entire period are reported in Table 2.

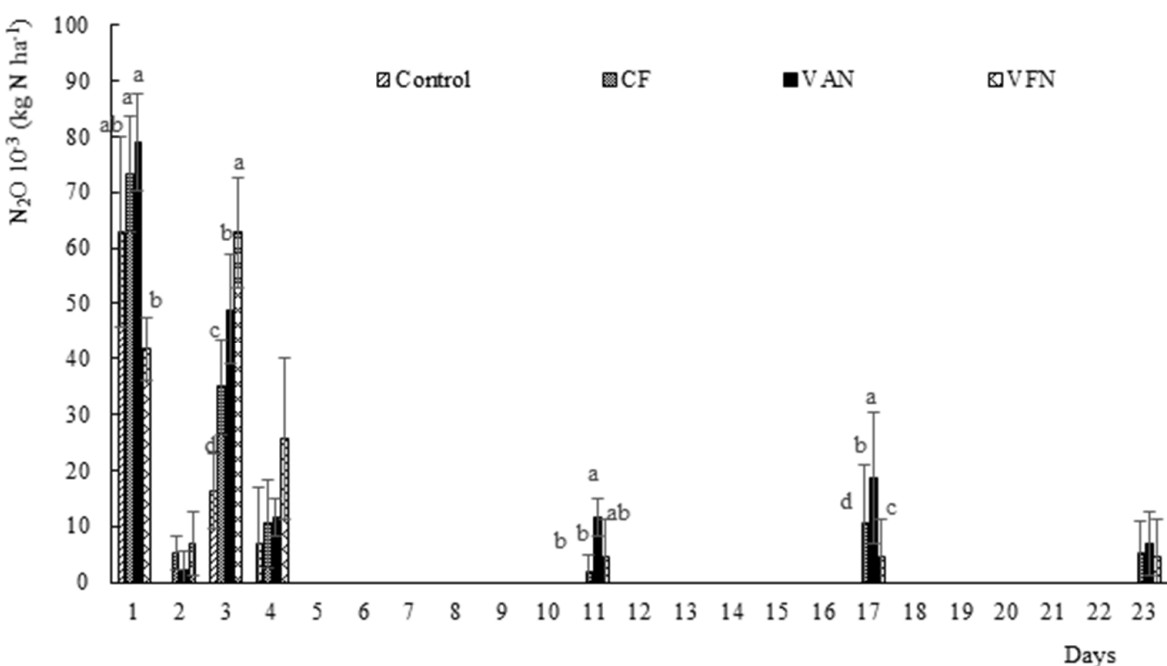

**Figure 4.** $N_2O$ emission dynamics ($N_2O$ $10^{-3}$ kg N ha$^{-1}$) for barley in season 2018/2019. Different letters indicate significant difference, according to Dunn's post-hoc test ($p < 0.05$), between treatments.

Despite the highest cumulative flux measurements reported for the VAN compared to the Control, the results were not significantly different (Table 2). Results showed that emission dynamics were not significantly influenced by fertilizer treatment, even if the lowest level was registered for the Control. Chen et al. [60] reported that N mineral fertilization increased the net nitrification rate and $N_2O$ release simultaneously, suggesting that $N_2O$ production was mainly due to ammonia oxidation. In this experiment, the highest emissions were registered after fertilization, but no significant variations were registered after the respective rainfall events. In particular, three days after fertilization, a $N_2O$ emission peak was observed in all treatments, corroborating previous observations by [61] and [52]. The variable results and the high standard mean error (Figure 3) were also reported by [62], where high coefficients of variation for $N_2O$ emissions after fertilization were reported. As $N_2O$ is produced from both denitrification and nitrification processes in soils, the heterogeneity in emissions might be due to a differing distribution of related microorganisms [63].

The low $N_2O$ emissions measured in our experiment were primarily because the barley crop consumed a relevant amount of N. In this sense, plant N uptake significantly reduced available N for denitrification with a consistent reduction in $N_2O$ emissions [11,64]. Moreover, the low temperatures during the monitoring period (Figure 1) significantly hampered $N_2O$ emission dynamics, as was also reported by [11].

*3.5. Crop Yield and N Content under Different Fertilizer Sources and Rates*

In N-fertilized plots, GYs were significantly higher compared to the Control, ranging from 5.45 to 6.86 t ha$^{-1}$ (Figure 5). The yield increases due to N fertilization were 23%, 55% and 40% in the CF, VAN and VFN treatments, respectively, compared to the Control (4.42 t ha$^{-1}$). There was no significant difference between the two variable-rate treatments (Figure 4), in which the same N amount was used, but in different forms. Significant differences were found between the two variable treatments and the Control for SB (Figure 5). GY and SB measurements for the Control were similar to those obtained in CF, where 150 kg N ha$^{-1}$ was supplied.

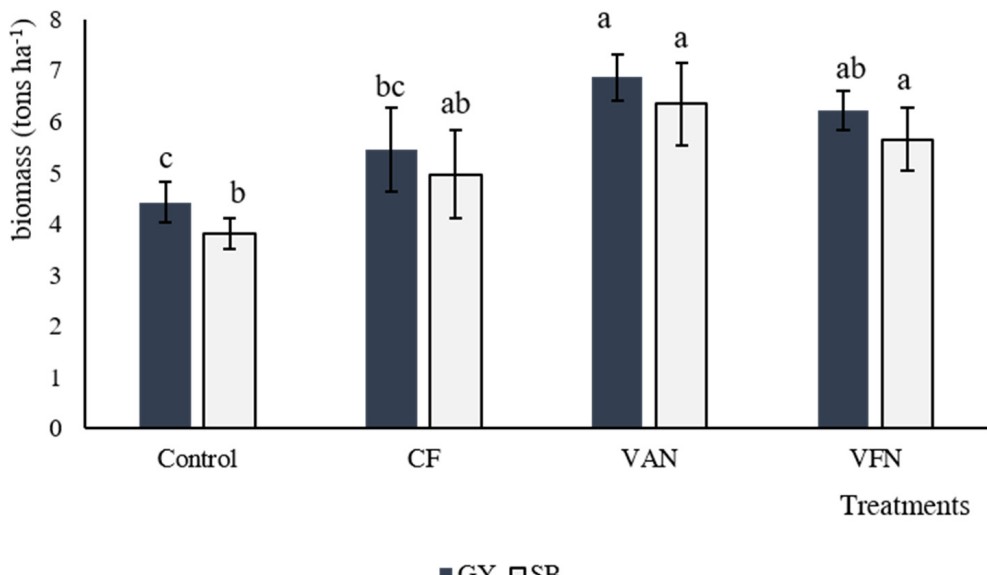

**Figure 5.** Effect of different N treatments on grain yield (GY) and straw biomass (SB) (t) of barley cultivated during the season 2018/2019. Control (0 kg N ha$^{-1}$); CF (conventional fertilization—150 kg N ha$^{-1}$); VAN (variable N application rate with ammonium nitrate—35 kg N ha$^{-1}$); VFN (variable N application rate with a foliar liquid N fertilizer—35 kg N ha$^{-1}$). Different letters indicate significant difference, according to Dunn's post-hoc test ($p > 0.05$), between treatments.

The variable-rate treatments produced the highest GY, even though the amount of fertilizer was five-fold less than the CF treatment. This is due to a higher efficiency of N utilization and initial soil conditions (Table 1). This result confirms the capacity of the crop sensor to assess the real N requirements from plant growth measurements in mid-season [65]. Colaco et al. (2018) [66] observed that using sensor-based technology for identifying the period of maximum crop demand, yields increased by 17% compared to conventional methods. This is in accordance with our previous study [67] in which we observed that performing fertilization at the right growing stage allows a reduction of 50% in the N rate while maintaining high yields of barley. The relevance of the rational use of fertilizers was highlighted by [68] who observed that on reducing the N rate on sweet potato by 20%, yields increased by 16.6–19% compared to conventional doses. In this study, yields are principally affected by a more efficient application rate, rather than the N source, and the initial soil N content. In addition, the similar crop productivity response obtained under both the Control and CF, supports the hypothesis that over-fertilization of barley in moderately N-rich soils did not increase crop yield.

Total N uptake of fertilized treatments was two-fold higher in the VAN and VFN compared to the Control (Table 3). Significant differences were observed between the CF, Control and variable-rate plots, respectively, according to the Tukey comparison tests (Table 3). However, differences were not observed between the VFN and VAN treatments. N indices were higher for variable-rate treatments (VAN and VFN) in comparison to that of the CF, for both N source and rate (Table 3).

It was previously reported that increasing the N rate reduces the N uptake efficiency and increases the probable loss of residual N [69]. The significantly higher N surplus of the CF treatment would potentially be either stored in soil or lost through leaching, thereby rendering the over-fertilization of the crop as an environmental issue of great concern [70].

**Table 3.** Effect of Treatments (Nrate + Source) and Nrate on crop N uptake (Nup), N surplus (Nsur), Physiological Efficiency (PE), Agronomic Efficiency (AE), partial factor productivity of applied nutrient (PFP), Recovery Efficiency (RE).

| | Treatment | Nup (kg ha$^{-1}$) | Nsur (kg ha$^{-1}$) | PE (kg kg$^{-1}$) | AE (kg kg$^{-1}$) | PFP (kg kg$^{-1}$) | RE (%) |
|---|---|---|---|---|---|---|---|
| Nrate + Source | Control | 76 ± 2.71 b | −76 ± 2.71 b | 57 ± 4.86 | - | - | - |
| | CF | 115 ± 30.56 b | 40 ± 21.94 a | 48 ± 6.31 | 8 ± 6.34 b | 36 ± 6.34 b | 0.26 ± 0.20 b |
| | VAN | 135 ± 9.70 a | −106 ± 21.21 b | 50 ± 1.59 | 81 ± 6.34 a | 228 ± 18.04 a | 2.01 ± 0.70 a |
| | VFN | 143 ± 16.08 a | −113 ± 21.92 b | 43 ± 5.35 | 60 ± 6.34 a | 207 ± 15.79 a | 2.26 ± 0.73 a |
| Nrate | 0 | 76 ± 2.71 c | −76 ± 2.71 b | 57 ± 4.86 | - | - | - |
| | 35 | 140 ± 19.72 a | −110 ± 19.72 b | 47 ± 5.75 | 70 ± 19.10 a | 218 ± 19.10 a | 2.13 ± 0.65 a |
| | 150 | 109 ± 21.94 b | 40 ± 21.94 a | 48 ± 6.31 | 8 ± 6.34 b | 36 ± 6.34 b | 0.26 ± 0.20 b |
| ANOVA | Nrate + Source | ** | *** | NS | *** | *** | ** |
| | Nrate | *** | *** | NS | *** | *** | *** |

Control (no nitrogen); CF (conventional fertilization); VAN (variable-rate fertilization with foliar liquid N fertilizer); VFN (variable-rate fertilization with ammonium nitrate). Values (mean ± SE) followed by different letters within each column are significantly different at the probability level of 0.05. ANOVA has been performed for Treatment (Nrate + Source) and Nrate—means no data. The symbols ** and *** indicate significant levels for $p < 0.05$ and $p < 0.01$, respectively

Based on the present results, we may conclude that the use of proximal sensors is an effective strategy to improve N uptake by crops. The lowest values for the N indicators observed under the CF treatment evidenced that the N supplied with traditional practices often exceeds the requirement of the crop. In general, the use of proximal sensors to manage N fertilization results in higher NUE in comparison to traditional practices, while maintaining similar yields [38]. Tubaña et al. (2008) [71] showed that midseason N application on corn, driven by remote sensors, could improve NUE. Similarly, the authors of [72] found that optical sensor-based N management in wheat significantly reduced the N application rates, enhanced N uptake and decreased the apparent N loss, without significant yield decreases. In our research, RE, AE and PFP all decreased with the increasing N rate, as was also reported by [73]. Furthermore, from the analysis of PE, no significant differences were observed between treatments, showing that there is a limit to N use by plants, above which there is no N uptake [74]. Our results indicated that the use of remote or proximal sensors contributes to a more efficient use of N compared to conventional fertilization strategies. Other studies showed that the use of sensor-based N management approaches permits the improvement of NUE and yield in comparison to conventional fertilization [38,71,75].

The results for grain Pc are reported in Figure 6. The highest Pc and Ng were found in the VFN (14% and 2.30%) followed by the CF (13% and 2.10%) and the VAN (12% and 1.98%) treatments, respectively. The Control showed the lowest amounts of Pc and Ng (10% and 1.74%), significantly different only from the VFN treatment. The Control, VAN and CF treatments did not significantly diverge, even though the applied N rates were significantly different.

Foliar fertilization may strongly influence root growth and soil N uptake [76], inducing a higher protein concentration in the grain. The rate and the source in VFN stimulated a significant production of Pc in comparison to the other treatments. Despite the low N rate of foliar application, barley under VFN produced the same Pc than CF, confirming the suitability of the variable-rate strategy to optimize N fertilization efficiency. Invariably, the efficacy was influenced by the fertilization time, supplied late during the crop growth. Foliar N applications are readily available for crops, due to the leaf absorption [77]. Moreover, N supplied later in the vegetative season may be more efficiently stored in the grains and less in the vegetative parts [78]. Accordingly, the authors of [79] reported that foliar N fertilization in post-pollination was able to enhance protein content approximately 70% of the time when the yield goal was exceeded in wheat and other crops. Instead, the authors of [80] reported that top-dress granular N fertilization produced a higher protein accumulation than foliar application on wheat. Nevertheless, both granular and foliar N application treatments provided higher results than the Control. Further research, related

to the influence of the interactions of both N rate and source on cereal Pc, using precision fertilization, is a requirement.

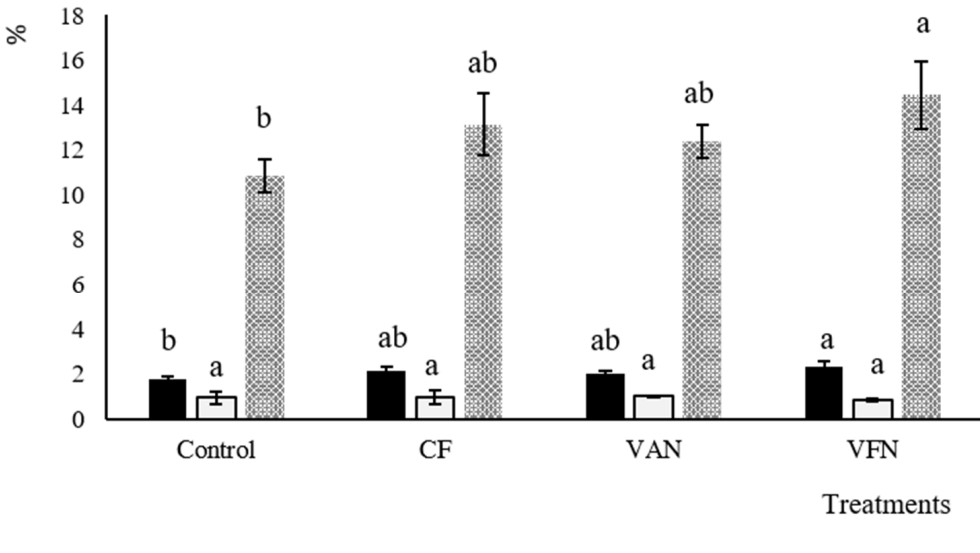

**Figure 6.** Effect of different N treatments on grain (Ng) and straw (Ns) N uptake and protein concentration (Pc) (%) of barley cultivated during the season 2018/2019. Bars with the same letter are not significantly different ($p > 0.05$). Control (0 kg N ha$^{-1}$); CF (conventional fertilization—150 kg N ha$^{-1}$); VAN (variable N application rate with ammonium nitrate—35 kg N ha$^{-1}$); VFN (variable N application rate with a foliar liquid N fertilizer—35 kg N ha$^{-1}$). Different letters indicate significant difference, according to Dunn's post-hoc test ($p < 0.05$), between treatments.

## 4. Conclusions

Precision fertilization is an effective way of optimizing the use of resources, as N, by synchronizing the crop requirements with the nutrients' supply, thereby reduces the environmental impacts of fertilization. If this is true for what concerns the impacts from the production process, due to the reduced amount of used fertilizers, direct GHGs emissions from the soil are highly dependent on site-specific pedoclimatic conditions. This preliminary study suggests that precision fertilization does not lead to a reduction in GHGs emission fluxes in the short term under the current experimental conditions. Despite using one-fifth of N in precision fertilization treatments (VAN and VFN) compared to conventional (CF), the cumulative emissions of the three considered gases ($CO_2$, $CH_4$ and $N_2O$) are similar. External weather conditions (temperature and precipitation trends) and intrinsic soil N content, probably outweighed fertilization thereby masking its effect.

The analysis of the N indicators confirms the role of precision fertilization to reduce the amount of fertilizers while maintaining high yields and increasing environmental performances.

This kind of study can provide crucial information on the understanding of GHGs emission dynamics in the short term under specific environmental conditions and crop growth stages. This will allow for the adjustment of fertilization towards a synchronization on crop requirements contributing to the sustainable development of agriculture.

**Author Contributions:** Conceptualization, C.F. and L.V.; methodology, C.F. and L.V.; formal analysis, C.F. and M.N.; investigation, C.F. and L.V; data curation, C.F. and M.N.; writing—original draft preparation, C.F. and L.V; writing—review and editing, A.D.M. and S.O.; supervision, A.D.M. and S.O.; project administration, S.O.; funding acquisition, A.D.M. All authors have read and agreed to the published version of the manuscript.

**Funding:** This research was funded by the project "PANE + DAYS" co-financed under Tuscany FEASR 2014-2020 Rural Development Programme; Measure 16.2; GO-PEI; The project was also supported by "Fondazione Cassa di Risparmio di Firenze" and "Fondazione per il Clima e la Sostenibilità".

**Institutional Review Board Statement:** Not applicable.

**Informed Consent Statement:** Not applicable.

**Acknowledgments:** The authors also wish to thank Roberto Vivoli from DAGRI for his support during the whole experiment.

**Conflicts of Interest:** The authors declare no conflict of interest. The funders had no role in the design of the study; in the collection, analyses, or interpretation of data; in the writing of the manuscript; or in the decision to publish the results.

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
