# Peer review of "Short-term Response of Greenhouse Gas Emissions from Precision Fertilization on Barley"

_agronomy, doi:10.3390/agronomy13010096_

Round 1

Reviewer 1 Report

The authors evaluated the responses of short-term GHGs emissions and different N use indicators to different precision fertilization strategies on barley, with aim to reduce environmental impacts of over-dose N application. This is a valuable and useful study. Overall the manuscript is clear and well-written and provides useful data for researchers involved in these systems. However, the ms still needs a major revision since I have the following concerns.

As stated in the Introduction part, methane and nitrous oxide have a Global Warming Potential (GWP) of 28 CO2eq and 265 CO2eq, repectively, and the three GHGs emissions were all observed in this study. Therefore, I suggest the GWP of CO2, CH4, N2O should be calculated to compare the comprehensive warming potential induced by different fertilization strategies. The related discussion should be focused on the short-time responses of comprehensive potential of the three gases rather than each single gas.

Another concern is about the measurement of CO2 emission. In general, CO2 production is mostly regulated by the interactions among soil respiration and plant photosynthesis. Regarding the method, The CO2 emission was measured using the static chamber method (P131-132). Is the chamber transparent or a dark one? The chamber type is important for the accuracy of CO2 meassurement. If it is transparent, the CO2 emssion must be influneced by plant photosynthesis instead of fertilization strategies differences. Therefore, more details should be declared in the method part, and the accuracy of CO2 measuremtns by the static chamber method should also be discussed in the main text. 

Methane is the end products of the degradation of organic matter under anaerobic, and methane production mainly occurs in flooded wetland soils, such as paddy rice fields. Regarding the CH4 emission in this study, I suggest the authors foucs on the emission after the rainfall, which might induce the anaerbobic environment and the activity of methanogens.  

P77-80 Based on the results and discussion parts the authors showed, the observation of the responses of short-term GHGs emission to different N fertilization strategies was only one of the aims of this study. The analysis of yeilds, biomass and different N use indicators should be addressed in this paragraph.

Author Response

1- As stated in the Introduction part, methane and nitrous oxide have a Global Warming Potential (GWP) of 28 CO2eq and 265 CO2eq, repectively, and the three GHGs emissions were all observed in this study. Therefore, I suggest the GWP of CO2, CH4, N2O should be calculated to compare the comprehensive warming potential induced by different fertilization strategies. The related discussion should be focused on the short-time responses of comprehensive potential of the three gases rather than each single gas.

Thanks for the suggestion. We included a brief description of GWP calculation in M&M (Line 146 - 151) and an additional chapter (3.5 Global Warming Potential assessment) in Results and Discussions (Line 322- 332).

2- Another concern is about the measurement of CO2 emission. In general, CO2 production is mostly regulated by the interactions among soil respiration and plant photosynthesis. Regarding the method, The CO2 emission was measured using the static chamber method (P131-132). Is the chamber transparent or a dark one? The chamber type is important for the accuracy of CO2 meassurement. If it is transparent, the CO2 emssion must be influneced by plant photosynthesis instead of fertilization strategies differences. Therefore, more details should be declared in the method part, and the accuracy of CO2 measuremtns by the static chamber method should also be discussed in the main text.

We agree with the reviewer about the importance of chamber type on soil CO2 fluxes. We used dark chambers and we included this specification into the text (Line 141). All details on methodology and chamber construction are included in reference 34 and 42.

3- Methane is the end products of the degradation of organic matter under anaerobic, and methane production mainly occurs in flooded wetland soils, such as paddy rice fields. Regarding the CH4 emission in this study, I suggest the authors foucs on the emission after the rainfall, which might induce the anaerbobic environment and the activity of methanogens. 

Thanks for the suggestion. The effect of precipitation on CH4 emissions was one of our first thought during data discussion. However, we observed no effect of precipitations on CH4 emissions dynamics. Even after the higher precipitation event (day 16 with 15 mm of precipitation) we noticed very few CH4 emissions. Nevertheless, at line 267 - 269 we mentioned that precipitation would played some role on CH4 fluxes dynamics.

4- P77-80 Based on the results and discussion parts the authors showed, the observation of the responses of short-term GHGs emission to different N fertilization strategies was only one of the aims of this study. The analysis of yeilds, biomass and different N use indicators should be addressed in this paragraph.

Thanks for the suggestion. We included a sentence about this part at line 78 – 80.

Reviewer 2 Report

Specific comments:

  1. Total N is not a physical parameter (table 1).
  2. Please complete table 1 with the pH value, the contents of organic carbon, and available forms of P and K (table 1).
  3. Please provide WRB soil classification.
  4. What about fertilization with other nutrients e.g. P and K?
  5. Please correct editing and language errors, e.g. CO2, CH4 and N2O (L. 135)
  6. Please add references to all equations used.
  7. Please provide methods and references for soil analysis.
  8. Please move subsection 3.1. Meteorological conditions during the study period to Materials and Methods.
  9. All abbreviations and acronyms used in tables and figures should be defined in the table notes or figure captions.
  10. Please remove unnecessary numbers of references (L. 61-617).

Author Response

  1. Total N is not a physical parameter (table 1).

Table caption corrected

  1. Please complete table 1 with the pH value, the contents of organic carbon, and available forms of P and K (table 1).

We included only pH and organic matter content as we don’t have data about P and K.

  1. Please provide WRB soil classification.

Thanks for the suggestion, we included this information at line 88.

  1. What about fertilization with other nutrients e.g. P and K?

This experiment was focused only on N fertilization and no other nutrients were provided to crops.

  1. Please correct editing and language errors, e.g. CO2, CH4 and N2O (L. 135)

Thanks, corrected

  1. Please add references to all equations used.

Thanks, we added the missing references (44, 45, 46, 47)

  1. Please provide methods and references for soil analysis.

We included references for soil analysis methodology (Line 108 -111 ref. 37, 38, 39)

  1. Please move subsection 3.1. Meteorological conditions during the study period to Materials and Methods.

Thanks for the suggestion. However, we’d like to maintain this section in chapter 3 since these are the first results we collected. In Materials and Methods we described the meteorological data we collected and we mentioned the weather station.

  1. All abbreviations and acronyms used in tables and figures should be defined in the table notes or figure captions.

Thanks, we corrected the missing expainations.

  1. Please remove unnecessary numbers of references (L. 61-617).

Thanks, corrected.

Reviewer 3 Report

The title is very clear and informative about the content of the manuscript.

 Soils that are depleted of nutrients and lack a sufficient supply of organic or chemical fertilizers will clearly be unable to meet the demands of high production. The handheld GreenSeeker crop sensor was well chosen because it can be used to make objective decisions about how much fertiliser to apply to the crop, resulting in more efficient fertiliser use that benefits both the bottom line and the environment.

 Introduction

The introduction provides enough background information about the topic to give the reader a quick understanding of the short-term GHG emission dynamics by comparing different N fertilisation strategies. The introduction includes a number of references (33), all of whom have conducted research in this field.

I think, the motivations for this study is clear stated at the end of the paragraph.

 Materials and Methods

The methods used are appropriate to the aims of the study, especially given the main focus of the paper is not to develop a novel technique, but to assess the specific barley’s N requirements through the GS measurements and the effect of different fertilization strategies in terms of yields and N uptake.

There is sufficient information provided by the authors, for a capable researcher to reproduce the experiments described.

I don’t think any additional experiments are necessary to validate the results presented here, because the results themselves are important and also the technique used to obtain these results is important; both are very well described.

Appropriate references are cited where previously established methods are used.

 Results and discussions

The results are clearly explained and presented in an appropriate format.

The figures and tables from the manuscript are easy to interpret. The figures (six) and tables (two) show essential data for the main GHG emissions dynamics in barley and the effect of different N treatments on barley grain yield and straw biomass. There are no data duplicated from graphics and tables in text.

Appropriate statistical methods have been used to test the significance of the results.

I believe that all possible interpretations of the data have been considered, consistent with the available data and the findings are properly described in the context of the published literature.

 Conclusions

The conclusions of the study are supported by appropriate evidence and emphasize that the analysis of the N indicators confirms the role of precision fertilization to reduce fertilizers amount while maintaining high yields and increase environmental performances in crops.

 References

The literature cited (80 references) is balanced and relevant to the study, assertions are substantiating with recent references.  

Author Response

We'd like to thak the reviewer 3 for appreaciating our work.